# NAFLD (MASLD)/NASH (MASH): Does It Bother to Label at All? A Comprehensive Narrative Review

**DOI:** 10.3390/ijms25158462

**Published:** 2024-08-02

**Authors:** Consolato M. Sergi

**Affiliations:** 1Department of Laboratory Medicine, University of Alberta, Edmonton, AB T6G 2B7, Canada; csergi@cheo.on.ca; Tel.: +1-613-737-7600 (ext. 2427); Fax: +1-613-738-4837; 2Children’s Hospital of Eastern Ontario, University of Ottawa, Ottawa, ON K1H 8L1, Canada

**Keywords:** liver, NAFLD, MASLD, NASH, MASH, nomenclature, taxonomy, classification, scores

## Abstract

Nonalcoholic fatty liver disease (NAFLD), or metabolic dysfunction-associated steatotic liver disease (MASLD), is a liver condition that is linked to overweight, obesity, diabetes mellitus, and metabolic syndrome. Nonalcoholic steatohepatitis (NASH), or metabolic dysfunction-associated steatohepatitis (MASH), is a form of NAFLD/MASLD that progresses over time. While steatosis is a prominent histological characteristic and recognizable grossly and microscopically, liver biopsies of individuals with NASH/MASH may exhibit several other abnormalities, such as mononuclear inflammation in the portal and lobular regions, hepatocellular damage characterized by ballooning and programmed cell death (apoptosis), misfolded hepatocytic protein inclusions (Mallory–Denk bodies, MDBs), megamitochondria as hyaline inclusions, and fibrosis. Ballooning hepatocellular damage remains the defining feature of NASH/MASH. The fibrosis pattern is characterized by the initial expression of perisinusoidal fibrosis (“chicken wire”) and fibrosis surrounding the central veins. Children may have an alternative form of progressive NAFLD/MASLD characterized by steatosis, inflammation, and fibrosis, mainly in Rappaport zone 1 of the liver acinus. To identify, synthesize, and analyze the scientific knowledge produced regarding the implications of using a score for evaluating NAFLD/MASLD in a comprehensive narrative review. The search for articles was conducted between 1 January 2000 and 31 December 2023, on the PubMed/MEDLINE, Scopus, Web of Science, and Cochrane databases. This search was complemented by a gray search, including internet browsers (e.g., Google) and textbooks. The following research question guided the study: “What are the basic data on using a score for evaluating NAFLD/MASLD?” All stages of the selection process were carried out by the single author. Of the 1783 articles found, 75 were included in the sample for analysis, which was implemented with an additional 25 articles from references and gray literature. The studies analyzed indicated the beneficial effects of scoring liver biopsies. Although similarity between alcoholic steatohepatitis (ASH) and NASH/MASH occurs, some patterns of hepatocellular damage seen in alcoholic disease of the liver do not happen in NASH/MASH, including cholestatic featuring steatohepatitis, alcoholic foamy degeneration, and sclerosing predominant hyaline necrosis. Generally, neutrophilic-rich cellular infiltrates, prominent hyaline inclusions and MDBs, cholestasis, and obvious pericellular sinusoidal fibrosis should favor the diagnosis of alcohol-induced hepatocellular injury over NASH/MASH. Multiple grading and staging methods are available for implementation in investigations and clinical trials, each possessing merits and drawbacks. The systems primarily used are the Brunt, the NASH CRN (NASH Clinical Research Network), and the SAF (steatosis, activity, and fibrosis) systems. Clinical investigations have utilized several approaches to link laboratory and demographic observations with histology findings with optimal platforms for clinical trials of rapidly commercialized drugs. It is promising that machine learning procedures (artificial intelligence) may be critical for developing new platforms to evaluate the benefits of current and future drug formulations.

## 1. Introduction

Nonalcoholic fatty liver disease (NAFLD), or metabolic dysfunction-associated fatty liver disease (MASLD), is a condition that affects the liver and is commonly seen in individuals with metabolic syndrome, overweight, obesity, type 2 diabetes mellitus (T2DM), or other conditions that lead to insulin resistance [1,2,3]. MASLD includes patients with hepatic steatosis and at least one of five cardiometabolic risk factors. These include (1) body mass index (BMI) ≥ 24 kg/m^2^ OR waist circumference (WC) > 90 cm (M) 80 cm (F), defined as the BMI subgroup; (2) fasting serum glucose ≥ 5.6 mmol/L OR 2 h post-load glucose levels ≥ 7.8 mmol/L OR HbA1c ≥ 5.7% OR diagnosis of DM OR treatment for DM, defined as the DM subgroup; (3) blood pressure ≥ 130/85 mmHg OR specific antihypertension treatment, defined as the hypertension (HT) subgroup; (4) plasma triglycerides (TGs) ≥ 1.70 mmol/L OR lipid-lowering treatment, defined as the TG subgroup; (5) plasma high-density lipoprotein (HDL) ≤ 1.0 (M) and ≤1.3 mmol/L (F) OR lipid-lowering treatment, defined as the HDL subgroup. Patients with hepatic steatosis who do not meet the cardiometabolic risk factors are diagnosed with cryptogenic steatotic liver disease (SLD) [4,5,6].

Figure 1 depicts the current criteria derived from Rinella et al.’s paper in *Ann Hepatol* of the current year (this figure can be reproduced under Creative Common License 4.0 for non-commercial uses) [5,6].

Although there is ongoing research on the non-invasive evaluation of patients with suspected NAFLD/MASLD, the most effective method for accurately determining the extent of liver disease is still a liver biopsy. A satisfactory needle procedure on the liver remains invasive but is usually tolerated in adults. Conversely, liver biopsy is not well endured in children despite the need to carry it out for staging [7,8,9,10]. In pediatrics, although it is essential to note that liver biopsy is not a completely safe procedure, it is beneficial for diagnostic purposes and determining the risk of potential developing neoplastic disease [9]. However, the pros and cons of doing it must be weighed wisely in every patient. Therefore, it should only be considered when accurate information about the diagnostic classification and severity of NAFLD/MASLD is necessary for clinical decisions. This is particularly relevant when there is still some uncertainty about the disease state even after non-invasive testing. For optimal assessment of the hepatocellular injury, the pathologist must receive adequate clinical and laboratory information about the patient.

The diagnostic and therapeutic benefits of percutaneous liver biopsy in pediatric liver disease have been widely recognized. In addition to histopathological examination, liver tissue obtained through biopsy can be utilized for immunocytochemical and electron microscopy investigations, microbiological culture, and biochemical analysis of hepatic enzyme activity [9]. Regrettably, liver biopsy is frequently not recommended due to the presence of coagulopathy or significant ascites in children who would greatly benefit from histological investigation. Transjugular liver biopsy may be a reliable and efficient alternative for adult and pediatric patients in such or similar situations [11]. The primary reasons for performing a transjugular liver biopsy are severe coagulopathy and massive ascites. In cases of moderate ascites, or when measuring the wedged hepatic vein pressure or obtaining a hepatic venogram is necessary, transjugular liver biopsy may be the preferred method [11].

Biochemical information must be integrated with histopathology to furnish comprehensive data to the clinician, the patient, and the family following a biopsy. Histopathology is crucial, and to achieve the objective indicated above, it is critical to make every effort to liaise with the interventional radiologist and acquire sufficient tissue during the biopsy procedure. While there are no definitive standards for an adequate biopsy, a practical minimum size is 1.5 cm, utilizing a 16- or 15-gauge needle. Our optimal number of cores at the Children’s Hospital of Eastern Ontario, Ottawa, ON, Canada, is three adequately long cores. Additional tissue is required when using thinner gauges or when the biopsy is fragmented. A specific protocol has been developed at the Children’s Hospital of Eastern Ontario, Ottawa, ON, Canada [9] (Figure 2). A number of liver cores inferior to three may yield a higher probability of being inaccurately classified or evaluated, which is a widely recognized issue in cases of chronic viral hepatitis [12,13]. This comprehensive narrative review will highlight the assessment of NAFLD/MASLD using liver biopsy, the fatty cell change or steatosis with or without inflammation, steatohepatitis, as well as grading and staging.

Figure 3a–d recapitulates some of the most critical hepatologic patterns of hepatocellular injury, which may be encountered during the evaluation of a liver biopsy for MASLD/MASH. Inflammation should be classified as acute or chronic hepatitis. Inflammation can be limited to the portal tracts, can involve the limiting plate (interface hepatitis), or can involve the hepatic lobulus. Degeneration should be listed as ballooning, foamy, or steatosis. Necrosis can be coagulative or lytic (hydropic). A Councilman body, often referred to as a Councilman hyaline body or apoptotic body, is a pink-stained globule composed of pieces of dying liver cells. In the end, the fragments are engulfed by macrophages or nearby parenchymal cells. They are more often located in the liver of patients experiencing acute viral hepatitis, yellow fever, and other viral diseases. Coagulative necrosis is characterized by hepatocytes that are mummified and poorly stained. Lytic necrosis refers to the process in which hepatocytes, which are liver cells, enlarge and eventually burst. Necrosis can be centrilobular, focal, piece-meal, bridging, sub-massive, and massive. Fibrosis or increased collagen can be located at portal, central, and bridging areas. Cirrhosis is defined by regenerative nodules surrounded by fibrosis. Steatosis is characterized by the presence of neutral fat, specifically triglycerides, in liver cells. This signals a malfunction in lipid metabolism or lipoprotein formation, or an abnormal accumulation of adipose or dietary lipids in the liver. Swelling or hydropic alteration occurs due to abnormalities in the functioning of cell membranes and/or mitochondria.

This comprehensive narrative review was carried out to identify, synthesize, and analyze the scientific knowledge produced regarding the implications of using a score for evaluating NAFLD/MAFLD, adopting a scoping review methodology to increase rigor to the search. The search for articles was conducted between 1 January 2000 and 31 December 2023, on the PubMed/MEDLINE, Scopus, Web of Science, and Cochrane databases. This search was complemented by a gray search, including internet browsers (e.g., Google) and textbooks. The following research question guided the study: “What are the basic data on using a score for evaluating NAFLD/MASLD?”.

## 2. Methods

### Project

This extensive narrative review was conducted utilizing a scoping review methodology. We implemented the scoping review methodology recommended by the Joanna Briggs Institute [14]. The review was organized based on the following stages: the review’s guiding question and objective were formulated, the search strategy was developed, the databases were searched, articles were selected based on a review of their titles and abstracts, scientific articles were chosen based on a thorough reading of their full text, the findings were summarized, and the results were presented and discussed in a narrative style. The objective was to identify, synthesize, and analyze the scientific knowledge produced regarding the implications of using a score for evaluating NAFLD/MASLD in a comprehensive narrative review. The search for articles was conducted between 1 January 2000 and 31 December 2023, on the PubMed/MEDLINE, Scopus, Web of Science, and Cochrane databases. This search was complemented by a gray search, including internet browsers (e.g., Google) and textbooks. The following research question guided the study: “What are the basic data on using a score for evaluating NAFLD/MASLD?” All stages of the selection process were carried out by the single author. The inclusion criteria were articles that contained the three PCC elements, that answered the research question, and that were written in English in the selected period. Articles written in other languages and that did not answer the guiding question were excluded. For the search, Descriptors in Health Sciences and Medical Subject Headings (MeSH) were used for each item of the strategy and its related terms. To combine descriptors, the AND, OR, and NOT Boolean operators were used. The PRISMA Extension for Scoping Reviews (PRISMA-ScR) guidelines were followed [15,16].

## 3. Results

Of the 1783 articles found, 75 were included in the sample for analysis, which was implemented with an additional 25 articles from references and gray literature. The studies analyzed indicated the beneficial effects of scoring liver biopsies. Although similarity between alcoholic steatohepatitis (ASH) and NASH/MASH occurs, some patterns of hepatocellular damage seen in alcoholic disease of the liver do not happen in NASH/MASH, including cholestatic featuring steatohepatitis, alcoholic foamy degeneration, and sclerosing predominant hyaline necrosis. Generally, neutrophilic-rich cellular infiltrates, prominent hyaline inclusions and MDBs, cholestasis, and obvious pericellular sinusoidal fibrosis should favor the diagnosis of alcohol-induced hepatocellular injury over NASH/MASH. Multiple grading and staging methods are available for implementation in investigations and clinical trials, each possessing merits and drawbacks. The systems primarily used are the Brunt, the NASH CRN (NASH Clinical Research Network), and the SAF (steatosis, activity, and fibrosis) systems. The quality or risk of bias of the studies were assessed using ad hoc methodology. This approach is consistent with the guidance of the scoping review pioneers [17,18]. All selected articles were relevant for the narrative style of this comprehensive review and discussion of the single aspects of liver steatosis and implications for clinicians, pathologists, and healthcare administrators. This review will also include aspects not merely related to liver histology, but that seem to be highly relevant to clinical practice contemporarily, e.g., non-invasive evaluation of steatosis and fibrosis, genetic prediction of NAFLD/MASLD, as well as the potential role of artificial intelligence and machine learning for the current and future diagnostic protocols of NAFLD/MASLD.

## 4. Non-Invasive Evaluation of Steatosis and Fibrosis

The diagnosis and measurement of hepatic steatosis without invasive procedures mostly depend on biomarkers or imaging techniques, such as ultrasound-based methods (specifically liver ultrasonography and controlled attenuation parameter (CAP)) or radiological approaches using computed tomography (CT) or magnetic resonance imaging (MRI). In the past ten years, researchers have created methods to evaluate hepatic steatosis utilizing MRI technology. Specifically, multiparametric MRI involves utilizing several quantitative characteristics simultaneously [19,20]. These approaches also enable the assessment of multiple aspects of liver pathology. These approaches encompass proton density fat fraction (PDFF), magnetic resonance elastography (MRE), modified LookLocker inversion recovery (MOLLI), and diffusion-weighted imaging (DWI). MRI is now the most precise and sensitive technique for measuring proton density fat fraction. It is easier and more feasible than magnetic resonance spectroscopy. However, its use has been mostly limited to research and clinical trials. Chemical-shift-encoded MRI (CSE-MRI)-based proton density fat-fraction (PDFF) methods have demonstrated potential and offer greater accuracy compared to CAP [19]. However, these methods are still costly and not feasible for many clinical scenarios. The LiverMultiScan utilizes MRI-based imaging software (LiverMultiScan|Perspectum, https://www.perspectum.com/our-products/livermultiscan (accessed on 29 July 2024)) to provide a precise measurement of liver fat content and an indirect evaluation of fibrosis. These technologies are currently in the process of development and may not always be easily accessible in clinical settings. Moreover, the evaluation of fibrosis or fat may be influenced by the presence of hepatic iron or inflammation. Unlike Fibroscan, these metrics are not readily accessible at the moment, although rapid methods are currently being developed [19]. A recent meta-analysis of the Controlled Attenuation Parameter (CAP) method for evaluating fat content determined that the CAP is capable of accurately identifying severe steatosis in patients with viral hepatitis. However, it is not able to accurately classify the severity of steatosis in patients with NAFLD/MASLD [21]. Due to the presence of radiation, CT scans are not commonly employed for the identification of fat. In general, the CAP approach, specifically using the M probe, is commonly employed. CT and normal ultrasound, although less sensitive, can nonetheless be useful in identifying unexpected instances of steatosis. Several blood markers and scores have been created and suggested as indicators of hepatic fat. These include the SteatoTest, Fatty Liver Index, Hepatic Steatosis Index, Lipid Accumulation Product, Index of NASH, and NAFLD Liver Fat Score [19]. The NITs are derived from many blood analytes, such as ALT, a2-macroglobulin, apolipoprotein A-1, haptoglobin, bilirubin, gamma-glutamyl transferase, total cholesterol, triglycerides, and glucose. Additionally, clinical parameters, including age, gender, and BMI, are considered. The SteatoTest has demonstrated superior accuracy when compared to histological testing. Recent research has examined the efficacy of Cytokeratin 18 (keratin 18) and microRNAs as prospective chemical biomarkers, which could be valuable in the future. Liver tests of the standard variety have limited diagnostic use [22].

The utilization of imaging techniques is progressively growing in order to ascertain the existence of fibrosis and cirrhosis. Numerous studies have been conducted to compare the sensitivity and specificity of various approaches. The conclusions are partially contingent upon the underlying liver condition. Ultrasound is not very effective in diagnosing or ruling out cirrhosis [23]. Transient elastography (TE) is increasingly utilized for the identification of fibrosis and cirrhosis. The evaluation of fibrosis has been significantly transformed by the extensive adoption of Fibroscan and, to a lesser degree, by the utilization of serological markers. Various techniques can be employed to evaluate fibrosis in NAFLD. The serological tests often employed include the Enhanced Liver Fibrosis (ELF) test, which is derived from the measurement of certain biomarkers. There are various biomarkers, including hyaluronic acid, amino-terminal pro-peptide of type III procollagen (PIIINP), and tissue inhibitor of metalloproteinase 1 (TIMP-1), in assessing the severity of NAFLD/MASLD [19]. These biomarkers may need to be evaluated in multicenter clinical investigations before having a definitive statement. The NAFLD Fibrosis Score takes into account factors such as age, body mass index, blood glucose, platelet count, albumin, and the ratio of aspartate aminotransferase (AST) to alanine aminotransferase (ALT), as well as the Fibrosis (FIB)-4 Score, which considers age, AST, ALT, and platelet count. Additional serological tests include APRI, Fibrotest, Forns Index, HepaScore, and the recently developed NIS4 algorithm [24]. Overall, these tests exhibit satisfactory specificity but demonstrate reduced sensitivity in detecting severe fibrosis and cirrhosis.

The utilization of imaging techniques is progressively growing in order to ascertain the existence of fibrosis and cirrhosis. Numerous studies have been conducted to compare the sensitivity and specificity of various approaches. The conclusions are partially contingent upon the underlying liver condition. Ultrasound is not very effective in diagnosing or ruling out cirrhosis [23]. Overall, TE seems to be increasingly utilized for the identification of fibrosis and cirrhosis. A recent study conducted on 145 Japanese patients examined various methods for evaluating NASH/MASH [25]. The methods included magnetic resonance elastography, multiparametric MRI with LiverMultiScan, vibration-controlled transient elastography, and 2D shear-wave elastography. The study found that MR liver fat and cT1 were the most effective individual measures, and when combined in multiparametric MRI, they provided the best overall non-invasive test for assessing nonalcoholic steatohepatitis. For diagnosing fibrosis ≥ 1, MRE worked best, and for assessment of steatosis ≥ 1, MR liver fat was the best. Overall, as concluded by Roccarina and colleagues [26], non-invasive assessment of liver fibrosis and portal hypertension is a validated tool for the diagnosis and follow-up of patients and the combination of transient elastography and platelet count for ruling out varices needing treatment in patients with compensated advanced chronic liver disease [26]. Elastography methods are more accurate than basic serum non-invasive testing but are limited by increasing rates of unreliability with growing fat [27]. Similarly, Patel and Sebastiani determined that although these tests are being used more frequently in clinical practice, they were not specifically created to accurately represent the ongoing development of fibrosis, distinguish between different stages of the disease, diagnose NASH/MASH, or track changes in fibrosis or disease activity over time due to natural progression or medical treatment [28]. Moreover, they asserted that non-invasive testing should be seen as supplementary to, rather than as a substitute for, liver biopsy. Transient elastography can sometimes provide inaccurate results, resulting in a misdiagnosis of advanced fibrosis. This can occur in cases of hepatic congestion, biliary obstruction, cholestasis, amyloidosis, inflammation, acute viral hepatitis, benign and malignant liver lesions, and during the period after eating [29].

C-labeled breath tests, aminopyrine breath, galactose breath, methacetin breath, and keto-isocaproic acid breath tests are mostly used to assess the prognosis of individuals with cirrhosis, rather than to diagnose or determine the cause of the condition. An alternative method that could be valuable in identifying and treating some liver illnesses is the analysis of volatile organic chemicals, such as erpinene, dimethyl sulfide, and D-limonene, in exhaled breath [30,31].

## 5. Assessment of NAFLD/MASLD Using Liver Biopsy

When a biopsy is conducted to assess suspected NAFLD, the primary aim of the pathologist is to determine the nature of the illness [12,13]. Thus, preserving tissue for histology is critical, but fresh-flash-frozen tissue may be handy in several cases. Recently, a protocol has been established at the Children’s Hospital of Eastern Ontario [9]. The liver biopsy is critical in diagnosing the patient’s category. At its most fundamental level, this entails determining the presence of fatty liver disease and assessing the coexistence of additional disease processes, such as genetic hemochromatosis or autoimmune hepatitis [32,33]. Table 1 shows the diseases and/or conditions that can coexist with NAFLD/MAFLD.

Kleiner has presented a comprehensive list of potential diagnoses while evaluating NAFLD [10]. These diseases include no evidence of fatty liver disease, steatosis, steatohepatitis, zone 1 borderline pattern, and cryptogenic fibrosis/cirrhosis. In steatohepatitis (NASH), there is a zone-3-centered injury pattern with fatty cell change or steatosis, inflammation, ballooning, and potential MDBs and/or fibrosis.

In contrast, in the zone 1 borderline pattern, there is a zone-1-centered injury with portal inflammation or “portitis”, portal tract-based fibrosis, and zone-1-localized steatosis [12,13,34]. The use of pathologist panels to increase efficiency and accuracy is critical [35]. On the milder side of the range, the biopsy may reveal such a small amount of fatty cell change that it is not justifiable to classify the condition as fatty liver disease. Other possible alterations may include minimal inflammation in the liver tissue, slight scarring, or mild iron build-up. However, there are no definite indications of steatohepatitis, such as conspicuous ballooning injury or the presence of MDBs [35,36,37,38].

Misfolded proteins have become more relevant in a range of disorders known as “protein aggregation diseases”, defined by aberrant proteins’ build-up. One of the most essential functions of a living cell is folding proteins into their three-dimensional, functionally active structures. Proteins prone to aggregation may be misfolded due to hereditary or toxic factors and end up in the cytoplasm, nucleus, or extracellular compartments. Numerous disorders can be identified morphologically by these inclusion bodies. These comprise, among other things, TDP-43 aggregates in amyotrophic lateral sclerosis, neuronal Lewy bodies in Parkinson’s illness, neurofibrillary tangles in Alzheimer’s disease, and MDBs in NASH and other chronic liver ailments. These inclusions exhibit a typical molecular composition and a disease-specific protein “backbone”, reflecting a shared pathogenesis. They all share ubiquitin (Ub) and the multifunctional stress and adaptor protein p62/Sequestosome-1 (p62) as constant constituents. Hepatocytic protein inclusions, or MDBs, are seen in several chronic liver illnesses, including hepatocellular carcinoma (HCC), idiopathic copper toxicosis, alcoholic or NASH, and chronic cholestasis. Chronic exposure to substances, such as griseofulvin or 3,5-diethoxycarbonyl-1,4-dihydrocollidine (DDC), can cause animals to develop MDBs.

Interestingly, MDBs in humans and mice are reversible; in ASH, they vanish when alcohol consumption is stopped, and in mice, they disappear when they recover on a regular diet following extended DDC intoxication. MDBs mainly comprise p62, ubiquitin, and the cytoskeletal intermediate filament proteins keratin 8 (K8) and keratin 18 (K18). Moreover, the MM120-1 antibody recognizes a high-molecular-weight component that they carry. This component is a particular marker of MDBs because it has only been found in human and mouse MDBs thus far. Together with these primary MDB components, several other proteins have also been identified as facultative MDB components. These proteins include the structural homolog of p62, known as Neighbor of BRCA1 (NBR1), and heat shock proteins 25 and 70 (Hsp25 and Hsp70). Keratins are trans-amidated, hyperphosphorylated, and partially degraded in MDBs.

Furthermore, it has been observed that keratins, specifically K8, undergo a conformational shift from a mostly α-helical to cross-β-sheet (amyloid-like) structure. Previous in vitro investigations on the production of MDBs demonstrated that p62 binds to misfolded, ubiquitinated keratins. According to earlier reports, p62 uses its UBA (ubiquitin-associated) domain to bind polyubiquitinated proteins, form them into sequestosomes, and transport them to pathways for degradation. These pathways include the autophagic machinery, which breaks down larger structures, such as protein aggregates and organelles, and the ubiquitin proteasomal system (UPS), which breaks down ubiquitinated and soluble proteins. Furthermore, p62 itself can assemble into intracellular hyaline bodies (IHBs) in the livers of patients suffering from HCC and idiopathic copper toxicosis. Transitional phases between keratin-positive MDBs and keratin-negative IHBs have been identified in certain liver disorders. This situation leads to the hypothesis that p62 aggregates could act as a matrix to allow “abnormal” keratins to eventually be incorporated, resulting in the creation of MDB. It is unclear if this process applies to the pathophysiology of MDB in ASH, NASH, or similar animal models [39].

Artificial intelligence (AI) has been critical to improve and optimize the assessment of fatty cell change [36,40,41,42]. Recently, AI-based measurement tools for scoring NASH histology have been presented at pathology conferences. AI-based NASH predictions for NASH Clinical Research Network (CRN) necroinflammation grades and fibrosis stages aligned with expert consensus scores and were reproducible. In a retrospective analysis of the ATLAS trial, previously unmet pathological endpoints were met when scored by the AI-based NASH algorithm alone [43]. The AI results seem encouraging, and AI can assist pathologists in the histologic review of NASH clinical trials. AI can reduce inter-rater variability and offer a more sensitive and reproducible measure of patient therapeutic responses. The biopsy results may indicate severe fibrosis, namely, bridging fibrosis or cirrhosis, without any or mild presence of steatosis, ballooning damage, or MDBs [10,44,45]. The fibrosis may still be caused by increasing NASH, but the distinctive characteristics are no longer evident, and the disease may reoccur following organ transplantation [46,47,48,49,50,51,52,53]. The biopsy should undergo meticulous scrutiny to identify any alternative causes of advanced liver disease, such as indications of persistent cholestatic illnesses, autoimmune hepatitis, or toxic/metabolic disorders. If other causes of long-term liver disease are ruled out, a suitable clinical history or previous biopsy indicating steatohepatitis can indicate that NAFLD is the cause. Most biopsies conducted for NAFLD/NASH can be classified into one of the three remaining groups, which will be further described in the subsequent sections.

## 6. Fatty Cell Change with or without Inflammation

Steatosis is a common occurrence in individuals diagnosed with NAFLD (MASLD)/NASH (MASH), which is accompanied by several aspects of hepatocellular injury and portal tract damage (Figure 4). It is essential to the disease’s diagnosis [12,36,37,54]. If biopsies do not exhibit diagnostic characteristics of either NASH or the zone 1 borderline pattern mentioned below, the primary and, occasionally, the sole histological result of importance will be steatosis. In an average adult patient, the steatosis will be predominantly located in zone 3 [9]. The entire hepatic acinus becomes affected with increased steatosis, resulting in a panacinar fatty look. In patients with severe fibrosis in a patient with NASH, the differentiation between the acinar zones is no longer apparent, and the distribution of steatosis may become uneven. The accumulation of large lipid droplets mainly characterizes the presence of steatosis in NAFLD/NASH, although the size of these droplets can vary significantly (Figure 4). Hepatocytes can have a solitary, sizable vacuole that occupies the whole cytoplasm, exerting pressure on the nucleus toward the cell membrane, like an adipocyte. The occurrence of hepatocytes resembling adipocytes likely resulted in the previous terminology for steatosis: fatty infiltration or fatty cell degeneration.

Furthermore, alongside hepatocytes with significantly giant vacuoles, there will also be hepatocytes containing one or more smaller vacuoles that do not wholly occupy the hepatocyte. This form of steatosis is not macro-vesicular fatty cell change, despite the vacuoles being smaller than the nucleus. It is more appropriately referred to as “small droplet”-predominant macro-vesicular steatosis [38,55]. The variability in the size of vacuoles and the frequent occurrence of vacuoles not filling the cytoplasm pose a barrier when attempting to estimate the severity of steatosis. Hepatocytes exhibiting genuine micro-vesicular steatosis can be observed in NAFLD/MASLD, although they are more commonly found in the advanced stages of the illness [56]. These cells can be observed individually or in tiny clusters, perhaps indistinguishable from the surrounding macro-vesicular steatosis. Occasionally, areas of micro-vesicular steatosis may have hepatocyte cytoplasmic hyaline inclusions that resemble megamitochondria histologically. These inclusions might have an oval or elongated shape and measure several microns in size.

There have been no reports of diffuse micro-vesicular steatosis, which is observed in conditions such as alcoholic or toxic damage occurring in NAFLD/MASLD or NASH/MASH. NAFLD/MASLD cases falling within this diagnostic group typically exhibit varying levels of inflammation. Although similarity between alcoholic steatohepatitis (ASH) and NASH/MASH occurs, some patterns of hepatocellular damage seen in alcoholic disease of the liver do not happen in NASH, including cholestatic featuring steatohepatitis, alcoholic foamy degeneration, and sclerosing predominant hyaline necrosis. Generally, neutrophilic-rich cellular infiltrates, prominent hyaline inclusions and MDBs, cholestasis, and obvious pericellular sinusoidal fibrosis should favor the diagnosis of alcohol-induced hepatocellular injury over NASH/MASH. Although it is possible to observe steatosis without inflammation, it is uncommon to find a biopsy from the NASH Clinical Research Network that shows no inflammation whatsoever [57]. Inflammation can be present in the parenchyma (lobular inflammation) or the portal sections (portal or periportal inflammation) of the liver, similar to other chronic liver disorders. The significance of lobular inflammation has been considered more significant than the different factors, and the assessment of lobular inflammation is included as a constituent of the overall activity in all the primary scoring systems employed in NAFLD. Lobular inflammatory foci consist of tiny aggregations of mononuclear cells, namely, lymphocytes and macrophages, that are around the size of one to two hepatocytes. They can form clusters within the sinusoid or infiltrate hepatocyte cords, often leading to the death of individual hepatocytes. Microgranulomas, which are small clusters of macrophages, frequently occur due to lobular inflammation [58]. Acidophil bodies, which are hepatocytes exhibiting programmed cell death, can be seen, and their quantity is directly related to the intensity of the disease. Portal inflammation, if present, is often moderate and characterized by a sparse infiltration of lymphocytes and macrophages. The limiting plate may be locally compromised (interface hepatitis). The intensity of the inflammation in the portal increases as the disease progresses to genuine steatohepatitis and fibrosis, and glycogenosis is a pretty typical finding in NAFLD (MASLD)/NASH (MASH) and is independently associated with cellular ballooning, but also associated with lower steatosis and lower fibrosis [12,13,34,59]. Some researchers have proposed that alterations, such as portal inflammation, may indicate the onset of fibrosis [60].

## 7. Steatohepatitis

Early investigations on NAFLD found that the presence of steatohepatitis was linked to a higher risk of death compared to NAFLD without the diagnostic criteria of steatohepatitis, and follow-up research has validated this finding [61,62], although the correlation mostly relies on the extent of fibrosis. NASH/MASH is additionally linked to a greater prevalence of diabetes mellitus and more severe obesity, as well as elevated levels of serum aminotransferases. Due to these factors, most ongoing clinical trials concentrate on the subset of NAFLD patients specifically diagnosed with NASH/MASH. The pathologist must employ precise criteria for diagnosing steatohepatitis. Cellular enlargement can be detected in other forms of liver damage. Still, in the case of NAFLD, there is a specific type of swelling indicative of the condition known as steatohepatitis. Ballooned hepatocytes exhibit significant enlargement compared to the non-steatotic hepatocytes and display irregularly clumped and stranded cytoplasm. MDBs are atypical clusters of keratin filaments that can be observed as dense, pinkish inclusions, typically located close to the nucleus of hepatocytes. In the first phases of the disease, ballooned hepatocytes are primarily seen in zone 3, adjacent to the major central veins. However, these hepatocytes can also be observed in other areas as fibrosis advances. The degree of hepatocyte ballooning ranges from cases with numerous significantly enlarged hepatocytes in each acinus to occasional balloon cells with alterations that are only slightly aberrant. Diagnostic significance should only be attributed to hepatocytes displaying persuasive and characteristic alterations in the absence of expertise in NASH/MASH evaluation by the pathologist. The presence of MDBs can be verified by employing immunohistochemical techniques that target ubiquitin or p62. Balloon cells can also be detected without keratin 8 and 18 when examined using immunohistochemistry. However, interpreting the stains might be challenging when the ballooned cells are not visible in regular histology. Almost all cases that have ballooning also display some degree of fibrosis. A mere 6–7% of patients diagnosed with confirmed steatohepatitis were found to be without. The Masson trichrome and Sirius Red stains are commonly utilized to evaluate fibrosis in cases of chronic liver disease. The extent of fibrosis may be underestimated if only the standard hematoxylin and eosin stain is employed. The fibrosis in NASH/MASH initiates with the accumulation of fragile collagen strands in the perisinusoidal region, typically near the enlarged hepatocytes. These deposits increase in thickness, making them visible without needing specific connective tissue stains. Periportal fibrosis is characterized by the elongation of tiny fibrous partitions that disrupt the boundary plate and enclose clusters of hepatocytes. Bridging fibrosis is characterized by the formation of fibrotic linkages between nearby blood vessels. These connections typically link prominent veins to portal areas due to the early onset of zone 3 fibrosis. As bridging fibrosis advances and regenerating nodules develop, the structural integrity of the liver is compromised, resulting in cirrhosis. Even in cirrhosis, fibrosis can persist and progress as fibrotic bands expand, and vascular shunts are formed. Fibrosis advancement might reduce the other histologic characteristics of steatohepatitis [57,63]. The presence of steatosis is reduced, and its distribution is more uneven. Identification of balloon cells may become increasingly challenging, as they can be located throughout the regenerative nodules. Immunostaining for MDBs can be highly beneficial in narrowing down the range of potential diagnoses. Diagnostic difficulty arises when certain characteristics of steatohepatitis are detected, but not all, even after a thorough analysis of numerous biopsy levels. The biopsy may exhibit steatosis, inflammation, and fibrosis in a suggestive arrangement, but no balloon cells or MDBs may be discerned. Alternatively, fibrosis may not be observed, and the balloon cells may exhibit poor formation. The term “borderline steatohepatitis” may be employed in certain instances [10], although a wide use of this term seems to be discouraged by several authors and hepatopathologists. If only advanced fibrosis is present, along with non-specific inflammation and limited steatosis, it may not be able to definitively attribute the injury to NAFLD/MASLD. Typically, these cases are assigned a diagnostic label describing them and a remark clarifying their limitations. The design that marks the boundary of zone 1 indicates that children and adolescents are susceptible to NAFLD/MASLD and NASH/MASH, which may lead to the progression of severe fibrosis [64]. Out of a group of 176 children from the United States, 14% exhibited bridging fibrosis, whereas none showed signs of cirrhosis [65]. Approximately 50% of this group had either borderline or definite NASH/MASH with the histological characteristics. Interestingly, 28% displayed a unique NAFLD/MASLD pattern rarely found in adults. This pattern was predominantly noticed in younger children, specifically those who were pre-puberal, under 12 years old [65,66,67]. Referred to as the zone 1 borderline pattern, this condition is defined by steatosis in either the zone 1 or panacinar distribution, together with periportal fibrosis or portal-to-portal bridging fibrosis. Steatohepatitis usually does not have well-defined balloon cells and does not exhibit MDBs. Similar to steatohepatitis, children who show the zone 1 pattern of NAFLD are significantly more prone to displaying characteristics of metabolic syndrome and having elevated body mass index values [66]. The precise progression of this injury pattern is not yet fully understood. Still, studies conducted on a group of individuals at a particular point in time indicate that it may combine with more common steatohepatitis when the affected children enter puberty [67].

## 8. Grading and Staging

Grading and staging are variably used in pathology. The stage concept is generally easier to comprehend between the two and typically holds greater therapeutic significance. Stage refers to the extent of an illness, indicating the progression of the disease from its initial state to its most advanced stage. In oncology, carcinoma is at a high stage when it has spread extensively to other body parts through metastasis. In chronic diseases, such as those affecting the liver, lung, and kidney, the term “end-stage” refers to the point at which the organ experiences functional failure. Fibrosis, characterized by its steady advancement to cirrhosis, is the predominant histological characteristic of chronic liver disease used to determine the stage. To the best of our knowledge, liver biopsy remains the most effective method in determining the stage of liver disease before the onset of cirrhosis. While it can also be used to diagnose cirrhosis, there are alternative non-invasive clinical assessments available to classify patients with highly advanced liver disease. Understanding grading might be challenging, but in pathology, grade relates to the disease’s activity level or aggressiveness during assessment. Apart from oncology, grading is often used for iron accumulation in the liver [32,33,68,69,70]. The histological features related to grade in NAFLD/MASLD include steatosis, inflammation, and ballooning damage.

Identifying the essential components of grade necessitates conducting longitudinal investigations that involve analyzing paired biopsies. There is insufficient data to determine if the many grading systems in use accurately assess the pertinent histological characteristics. An active field of pathology research involves elucidating the precise combination of traits that are most strongly connected with grade. The process of grading and staging has practical consequences for the clinical management of patients. Stage typically holds more excellent predictive value for long-term prognosis than grade, while grade also holds significance since it indicates the urgency, or lack thereof, for treatment. The treatment strategies can vary significantly between a high-stage, low-grade disease and a low-stage, high-grade disease. The upcoming sections will examine the three most often utilized grading and staging systems for NAFLD/NASH: the Brunt system, the NASH CRN system, and the SAF (steatosis, activity, and fibrosis) system (Figure 5).

## 9. The Brunt System

In 1999, Dr. Brunt and colleagues introduced a grading and staging system for NASH [71]. The study’s objective was to develop and assess recommendations for the grading and staging of NASH, similar to the grading and staging methods currently established for chronic hepatitis. A comprehensive assessment of histological characteristics was conducted during the unbiased examination of 51 patients. The assessments involved evaluating the levels of steatosis, ballooning, lobular and portal inflammation, MDBs, acidophil bodies, PAS-positive Kupffer cells, glycogenated nuclei, and lipogranulomas using a semi-quantitative approach. The degree of fibrosis was assessed using a semi-quantitative method in three specific areas: perisinusoidal, portal, and bridging. In addition, the researchers also documented the existence or absence of other characteristics, as well as the spatial distribution of the observations. The assessment resulted in broad categorizations of the overall severity of NASH/MASH, which may be classified as mild, moderate, and marked based on the comparative severity of steatosis, ballooning, lobular, and portal inflammation. The original study excluded cases of NAFLD that were not classified as NASH. The approach was initially designed for steatohepatitis and no other forms of fatty liver disease. The fibrosis data were categorized into a staging scheme consisting of five clearly defined stages. A correlation between the severity of cases classed as mild, moderate, or significant steatohepatitis and the aminotransferase levels was observed by Brunt and colleagues. This study was the first to introduce a technique for classifying and categorizing steatohepatitis, and it has been effectively utilized in other clinical investigations of NASH. The method has limitations regarding unclear definitions for specific scores, and its applicability is restricted to cases classified as steatohepatitis. Due to the exclusion of pediatric patients, the system failed to address specific atypical symptoms of pediatric fatty liver disease. The NASH CRN system overcame several problems and can be seen as an intellectual successor to the Brunt system.

## 10. CRN System

The National Institute of Diabetes and Digestive and Kidney Diseases (NIDDK) created a clinical network in 2003 to investigate NASH’s pathogenesis, natural progression, and therapy [72]. The pathology committee was tasked with developing a scoring system to record all the histological characteristics commonly observed in clinical studies of fatty liver disease in juvenile and adult patients. The initial assessment involved assigning a numerical score to measure the extent of steatosis, lobular inflammation, ballooning, and fibrosis. A binary scoring system was also used to evaluate additional characteristics [73]. Before evaluating the initial cases from the network, the portal inflammation scale was extended to a range of 0 to 2. The approach incorporated many components, such as evaluating the regional distribution of steatosis and isolated portal fibrosis, intending to address unique pathological findings observed in pediatric fatty liver disease [64]. Following an anonymous review of 50 cases, encompassing pediatric and adult cases covering the entire range of histological variations in NAFLD/MASLD, a multivariable analysis was conducted to determine the traits most strongly linked to the diagnosis of NASH. The observed conditions encompass steatosis, lobular inflammation, ballooning, and fibrosis. The initial three elements were combined to yield a comprehensive score indicating the severity of histological changes, known as the NAFLD Activity Score (NAS). Fibrosis was designated as the stage representative. While the NAS was closely linked to the diagnosis, the authors did not intend for the score to be equivalent to the diagnosis. The diagnosis was nevertheless assessed using the criteria described above. A thorough examination of the data obtained from a study including 976 individuals revealed that both the NAS and the diagnosis had separate correlations with aminotransferase levels. However, only the diagnostic showed an independent connection with diabetes mellitus [74]. The NASH CRN system offers comprehensive capabilities for analyzing histology in juvenile and adult populations. It has been widely utilized in therapeutic trials and various research endeavors. The existing method has a limitation where the NAS assigns a lower importance to ballooning (up to 2 points) compared to lobular inflammation or steatosis (up to 3 points each). Additionally, specific research outside the network mistakenly equates the NAS with the diagnostic classification. Ongoing research is being conducted to widen the ballooning scale and identify the histological findings that are most indicative of fibrosis advancement. Bedossa et al. introduced a third approach to grading and staging fatty liver disease as part of the Fatty Liver Inhibition and Progression (FLIP) Consortium [75]. This approach aimed to develop a straightforward grading and staging system that could also be utilized to categorize instances for diagnosis. The initial group consisted of people who underwent bariatric surgery. The histological evaluation was restricted to a semi-quantitative scoring system for assessing the presence of steatosis, lobular inflammation, ballooning, and fibrosis. A composite activity score was calculated by combining lobular inflammation and ballooning values, while excluding steatosis. This decision was made since the extent of steatosis appeared to have a weaker association with the advancement of fibrosis than the other two parameters. The scores for steatosis, lobular inflammation, and ballooning are instantly converted into a diagnostic category, eliminating the pathologist’s need to evaluate the diagnosis individually. Based on this methodology, the diagnosis of steatohepatitis is made in biopsies that show any level of steatosis exceeding the 5% limit, as well as a positive indication of both lobular inflammation and ballooning. The three scores for steatosis (S), activity (A), and fibrosis (F) are consolidated into a unified alphanumeric code that provides a concise summary of the significant histological observations. The authors successfully proved the high level of consistency and accuracy in the results obtained by expert hepatopathologists and ordinary pathologists trained to utilize the system. The SAF method is more straightforward to implement than the other two systems and mitigates potential discrepancies among observers when determining the diagnosis [76]. It has limitations, as it does not include an evaluation of portal inflammation and does not consider the specific characteristics of fatty liver disease in children. The inflexible correlation between score and diagnosis may result in atypical classifications. However, due to the similarity in the scoring structure with the NASH CRN system, the SAF system will probably be valuable for tracking changes in randomized clinical trials.

## 11. Genetic Prediction of NAFLD/MASLD

Anstee et al. [77,78] provided an outstanding review, summarizing the results of candidate genes in a genome-wide association study (GWAS) on the phenotypes of NAFLD/MASLD. GWAS is an investigation of a comprehensive collection of genetic variations across the entire genome in several individuals to determine if any variation is linked to a specific characteristic. The review presented the findings in a tabular format, listing all the genes that may contribute to the differences in the progression and outcomes of nonalcoholic and alcohol-related liver diseases [79]. It provided a concise overview to depict the findings of the genetic studies and their connections to different sub-phenotypes of fatty liver diseases using “tree diagrams”. These diagrams utilize a hierarchical system of disease classifications and represent the chronological progression of the disease, from fatty liver to steatohepatitis and cirrhosis. This method utilizes diagnostic hierarchies to demonstrate the common genomic structure underpinning [79]. Adiponutrin is an enzyme that is produced by the *PNPLA3* gene. It is a member of the patatin-like phospholipase-domain-containing family and consists of 481 amino acids. Adiponutrin is also known as calcium-independent phospholipase A2ε. The genesis of this domain can be traced back to lipid hydrolases found in potatoes. It was named after the protein patatin, which is the most abundant in potato tubers. *PNPLA3* is mostly expressed in the liver, retina, skin, and adipose tissue [80]. The *PNPLA3* gene variant p.I148M (c.444C>G, rs738409) has been identified as the primary genetic factor determining the severity of fatty liver disease in both pediatric and adult patients, based on a series of important genetic studies [81]. Additional genes comprise *TM6SF2* and *MBOAT7*, in particular the TM6SF2 variant p.E167K (c.449C>T, rs58542926) as a factor that increases susceptibility to NAFLD/MASLD [82].

The *TM6SF2* variant has a lower frequency (7%) in Europeans compared to the *PNPLA3* p.I148M polymorphism, which is more common in all ethnic groups (23% in Europeans and 55% in Hispanics). The *TM6SF2* gene contains the genetic instructions for producing a protein consisting of 351 amino acids. This protein is anticipated to have 7–10 transmembrane domains. TM6SF2 is found in the endoplasmic reticulum and the Golgi complex of hepatocytes and enterocytes, where apolipoprotein-B-containing lipoproteins are synthesized [83].

A GWAS found that the rs641738 variant, located in the 30 untranslated region of the gene encoding membrane-bound O-acyltransferase-domain-containing 7 (*MBOAT7*), specifically lysophosphatidylinositol-acyltransferase 1, is associated with an elevated risk of alcoholic cirrhosis. In this study, 942,736 patients from the Dallas Heart Study underwent proton magnetic resonance spectroscopy to quantify intrahepatic triglycerides. Additionally, 1149 European individuals from the liver biopsy cross-sectional cohort were genotyped for rs641738.95. The rs641738 variant (g.54173068 C>T/c.50 G>A) leads to the production of p.G17E in the transmembrane channel-like 4. This variant is linked to the inhibition of MBOAT7 at both the messenger RNA and protein levels. The presence of the variant was also linked to higher levels of fat in the liver, more severe liver damage, and elevated stages of fibrosis compared to individuals who did not have the variant. Moreover, it could potentially increase the likelihood of developing hepatocellular carcinoma (HCC) in people who do not have cirrhosis. GWAS and its meta-analyses have discovered supplementary genetic variations linked to NAFLD. Speliotes et al. [84] found that genetic variations in or near glucokinase regulatory protein (GCKR), lysophospholipase-like 1 (LYPLAL1), and phosphatase 1-regulatory subunit 3b (PPP1R3B) are significantly linked to liver fat content and/or histopathologic NAFLD/MASLD phenotypes at the genome-wide level of significance. A comprehensive meta-analysis conducted on hemochromatosis, with a total of 66,000 cases and 226,000 controls from Europe and China, revealed that individuals who are homozygous carriers of the *HFE* variation p.282Y (rs1800562) have a ten-fold higher risk of developing NASH. Unlike HFE, the simple presence of α1-antitrypsin mutations encoded by the *SERPINA1* gene, even in a heterozygous state, significantly raises the likelihood of developing chronic liver disease. AATD is one important cause of liver cirrhosis in children and adults with M-like variants as well as Pi*Z variations, with no apparent association with abnormality of the *HFE* gene [33,85,86,87]. Homozygous refers to an individual having two identical copies of a particular gene. Individuals that possess the Pi*Z variation (p.E342K) are not simply carriers. Individuals with this genetic variant are at a significantly increased risk of developing both lung emphysema and liver cirrhosis. This is because the variant protein has a toxic effect on liver cells, leading to a gain-of-function toxicity. A study showed that the Pi*Z variation was linked to greater fibrosis stages in 1184 patients with NAFLD/MASLD, with an odds ratio of 2.3 [88]. According to Abul-Husn et al. [89], the mutation rs72613567:TA of *HSD17B13* is also linked to a decreased likelihood of developing chronic liver disease and progressing from steatosis to steatohepatitis. Moreover, after studying 5770 patients with cirrhosis and 572,850 control subjects from 7 different groups, researchers discovered that a specific genetic variation called p.A165T in the *MARC1* gene is linked to a reduced risk of cirrhosis (OR, 0.88; *p* = 2.1 × 10^−8^). This variation is also associated with lower levels of fat in the liver, as observed through computer tomography scans, and a decreased likelihood of being diagnosed with fatty liver by a physician with major academic burden [90,91,92,93,94,95]. Consistent with these results, the beneficial variant was linked to decreased levels of alanine transaminase and alkaline phosphatase in the blood, as well as lower total and LDL cholesterol levels. In addition, individuals who possess uncommon protein-truncating variations in the *MARC1* gene exhibited a decreased likelihood of liver disease, decreased hepatic enzyme activity, and lower levels of cholesterol. This provides further evidence that a defect in *MARC1* offers protection against cirrhosis. The precise method by which *MARC1* may influence liver damage and cirrhosis remains uncertain, but it is known that *MARC1* encodes a molybdenum-containing nitric oxide synthase that reduces the physiological substrate N(omega)-hydroxy-L-arginine and activates it. This enzyme is known as the mitochondrial amidoxime-reducing component.

## 12. Artificial Intelligence

Liver illnesses impose a substantial cost on global public health. Despite significant progress in recent years, there remain numerous obstacles in the identification and management of liver illnesses. In recent years, artificial intelligence (AI) has been extensively employed for diagnosing, categorizing the risk, and predicting the prognosis of different diseases using clinical datasets and medical pictures. Cumulative research has consistently demonstrated the effectiveness of this diagnostic method in identifying patients with NAFLD/MASLD and liver fibrosis, as well as assessing the severity of these conditions. Additionally, it has proven valuable in predicting treatment responses and the likelihood of HCC recurrence, as well as evaluating outcomes for liver transplantation recipients and the risk of drug-induced liver injury. Utilizing AI, digital pathology has the capability to accurately measure histological observations in a consistent manner [96,97,98,99]. Ratziu et al. [100] examined the assessment of histological characteristics of NASH by both pathologists and a machine learning (ML) pathology model. This post hoc analysis included data from a specific group of patients (n = 251) who had biopsy-confirmed NASH with fibrosis grade F1–F3. The data were obtained from a 72-week randomized, placebo-controlled trial of once-daily subcutaneous semaglutide at doses of 0.1, 0.2, or 0.4 mg (NCT02970942). Two pathologists examined the biopsies at the beginning and in the 72nd week. PathAI’s NASH ML models were used to analyze digitized biopsy slides and measure the extent of fibrosis, steatosis, inflammation, and hepatocyte ballooning. This was performed through categorical assessments and continuous scores. The pathologist and machine-learning-based categorical assessments found that a significantly higher proportion of patients achieved the primary outcome of NASH remission without worsening of fibrosis with semaglutide 0.4 mg compared to the placebo (pathologist: 58.5% vs. 22.0%, *p* < 0.0001; machine learning: 36.9% vs. 11.9%, *p* = 0.0015). Both methodologies identified a greater, albeit statistically insignificant, proportion of patients receiving semaglutide 0.4 mg compared to the placebo who achieved the secondary outcome of liver fibrosis improvement without worsening of NASH/MASH. ML analysis revealed that semaglutide 0.4 mg led to a substantial reduction in fibrosis compared to the placebo (*p* = 0.0099). This effect was found by ML continuous scores, but it was not observed using traditional pathologist or ML categorical assessment methods. The study concluded that ML categorical evaluations accurately replicated the findings of pathologists about the improvement of histological conditions, such as steatosis and disease activity, with the use of semaglutide. Continuous scores derived from machine learning techniques showed an antifibrotic impact that was not detected by traditional histopathology methods.

AI has been critical to improve and optimize the assessment of fatty cell change [36,40,41,42]. The AI-powered NASH predictions for NASH Clinical Research Network (CRN) accurately matched the expert consensus ratings for necroinflammation grades and fibrosis stages, and the results were consistent and replicable. Upon doing a retrospective study of the ATLAS experiment, it was found that the AI-based NASH algorithm successfully achieved previously unmet pathological endpoints when evaluated independently [43]. The AI results seem encouraging, and AI can assist pathologists in the histologic review of NASH/MASH clinical trials. AI can reduce inter-rater variability and offer a more sensitive and reproducible measure of patient therapeutic responses. The biopsy results may indicate severe fibrosis, namely, bridging fibrosis or cirrhosis, without any or mild presence of steatosis, ballooning damage, or MDBs [10,44,45]. The fibrosis may still be caused by increasing NASH/MASH, but the distinctive characteristics are no longer evident, and the disease may reoccur following organ transplantation [46,47,48,49,50,51,52,53]. The biopsy should undergo meticulous scrutiny to identify any alternative causes of advanced liver disease, such as indications of persistent cholestatic illnesses, autoimmune hepatitis, or toxic/metabolic disorders. If other causes of long-term liver disease are ruled out, a suitable clinical history or previous biopsy indicating steatohepatitis can indicate that NAFLD is the cause. Most biopsies conducted for NAFLD/NASH can be classified into one of the three remaining groups, as discussed above.

Currently, AI is extensively utilized in medical research, particularly in the field of imaging diagnosis. There is ample scope for enhancing algorithms pertaining to NASH/MASH diagnosis, encompassing relational analysis, quantitative (statistical) analysis, and hypothesis testing, among other possibilities. In addition, the advancement of AI is dependent on the involvement of medical institutions, medical professionals, academic organizations, companies, and third-party operators. In the future, artificial intelligence holds the potential to enhance our ability to detect patients with NASH and those at risk for progressive fibrosis. This can be achieved by objectively analyzing liver images and addressing the limitations in the histological examination of the liver. AI will be used in clinical practice to assist in the management and monitoring of liver-related illnesses. With the use of bigger groups of patients, it is highly probable that a diagnostic system for NASH/MASH using AI will be created and implemented in real-world medical settings.

The efficacy of deep learning is intricately tied to the abundance of data, indicating that inadequate training data can significantly impair the performance of deep learning models. Nevertheless, a significant challenge in utilizing deep learning for medical picture analysis is the absence of efficient annotation. Restricted labels lead to a scarcity of accessible data, hence posing challenges in effectively training deep learning models and giving rise to overfitting issues. Thus, transfer learning proves to be a viable solution in this particular scenario [101]. Unlike general deep learning algorithms that solve individual tasks, transfer learning aims to transfer acquired knowledge from a source task and utilize it to enhance learning in a target task. When it comes to transferring deep learning models, two commonly used procedures include fine-tuning and feature extraction [102]. Fine-tuning involves training a pre-trained model, derived from the source dataset, on the target dataset in order to adjust and optimize all parameters in the learnable layers of the networks. The feature extraction process maintains consistent settings throughout all layers, except for the top layer. The uppermost layer is linked to the classifier and is directly associated with the classification task. Finally, the ongoing surge in AI development is accompanied by persistent demands for applied ethics, aimed at harnessing the transformative capabilities of emerging AI technology. Consequently, a comprehensive set of ethical norms has emerged in recent years, encompassing concepts that technology creators should strive to follow to the greatest extent possible [103].

## 13. Conclusive Remarks

NAFLD (MASLD)/NASH (MASH): does it bother to label at all? Yes, indeed, it does, and the classification will determine the prognosis and outcome of patients and clinical trials. However, all current scores have some limitations, mainly due to the liver biopsy size, which may not fully mirror the clinical situation. We hope that additional parameters may be incorporated in the future to realize an effective and productive scoring system. An example may be presented from the Shimada system, which is used for the diagnosis, prognosis, and therapeutic response in patients with neuroblastoma, a pediatric mesenchymal malignant tumor, which has been investigated using fractal dimension, entropy, and lacunarity [104]. The histological alterations in NAFLD/MASLD are more varied than those observed in chronic hepatitis. This is because NAFLD/MASLD involves both the accumulation of fat in liver cells (steatosis) and a distinct form of injury to hepatocytes, in addition to inflammation and fibrosis. Steatosis can occur as the sole histological observation or is accompanied by varying lobular or portal inflammation levels. A diagnosis of steatohepatitis can be made when ballooning damage is present. Ballooning injury in steatohepatitis is typically always associated with inflammation and steatosis and is commonly accompanied by fibrosis. The initial histological alterations manifest mostly in zone 3 and are typically more pronounced. This distinction is another characteristic that sets apart NAFLD/MASLD patterns from chronic hepatitis. The only case that does not follow this norm is the zone 1 borderline pattern of NAFLD/MASLD, primarily observed in young children. Various methodologies have been created to evaluate the extent of fibrosis (stage) and other histological characteristics (grade) of NAFLD/MASLD and NASH/MASH in a systematic manner. These systems have been utilized in clinical investigations, including pharmaceutical trials, to establish a connection between histology findings and clinical data and monitor changes over time. Collaboration between pathologists and hepatologists is crucial in choosing an appropriate grading and staging system and accurately interpreting the findings from liver biopsies to guide patient treatment.

## Figures and Tables

**Figure 1 ijms-25-08462-f001:**
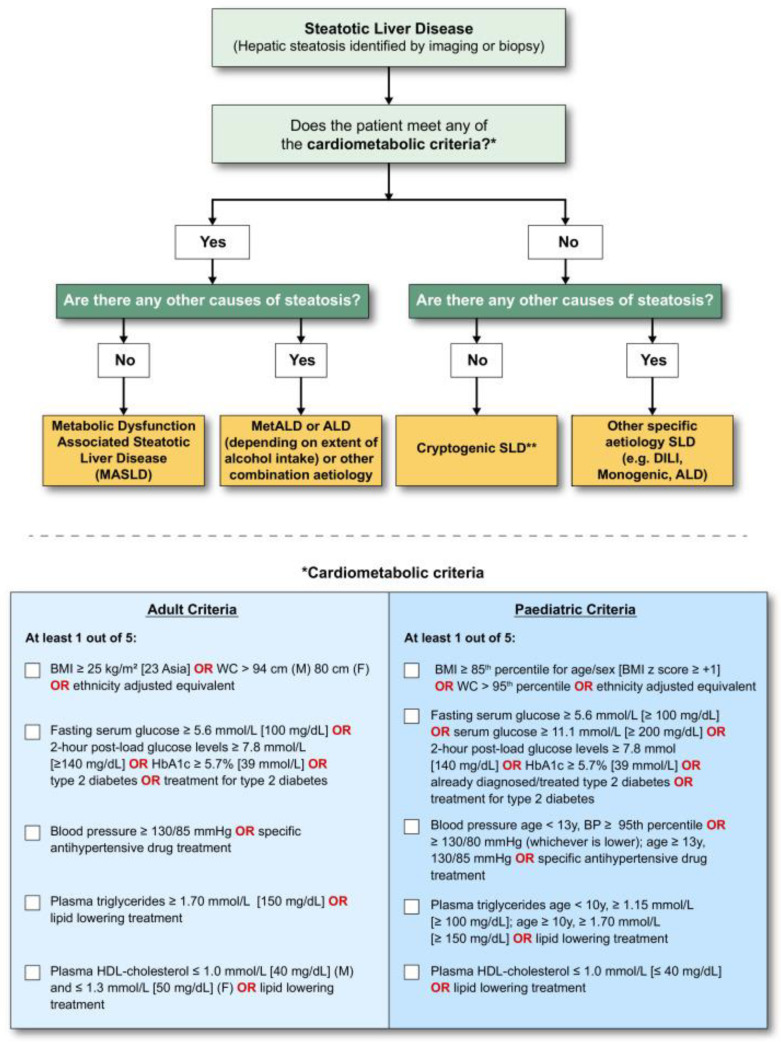
Diagnostic criteria for MASLD. If hepatic steatosis is present, the identification of any CMRF (*) would lead to a diagnosis of MASLD, provided that there are no other causes of hepatic steatosis. If other causes of steatosis are discovered, then this is in line with a mix of factors contributing to the condition. When it comes to alcohol, this is referred to as MetALD. If there are no obvious cardiometabolic criteria present, other potential causes must be ruled out. If no other cause is found, this is referred to as cryptogenic SLD (**). However, based on clinical opinion, it might also be seen as probable MASLD and would, therefore, require periodic examination on an individual basis. In cases of severe fibrosis/cirrhosis, the presence of steatosis may not be evident. Therefore, clinical judgment should be used, taking into consideration the patient’s clinical, metabolic, and risk factors, as well as ruling out other possible causes. Abbreviations: ALD, alcohol-associated/related liver disease; BMI, body mass index; BP, blood pressure; CMRF, cardiometabolic risk factors; DILI, drug-induced liver disease; MetALD, metabolic dysfunction and alcohol-associated steatotic liver disease; SLD, steatotic liver disease; WC, waist circumference.

**Figure 2 ijms-25-08462-f002:**
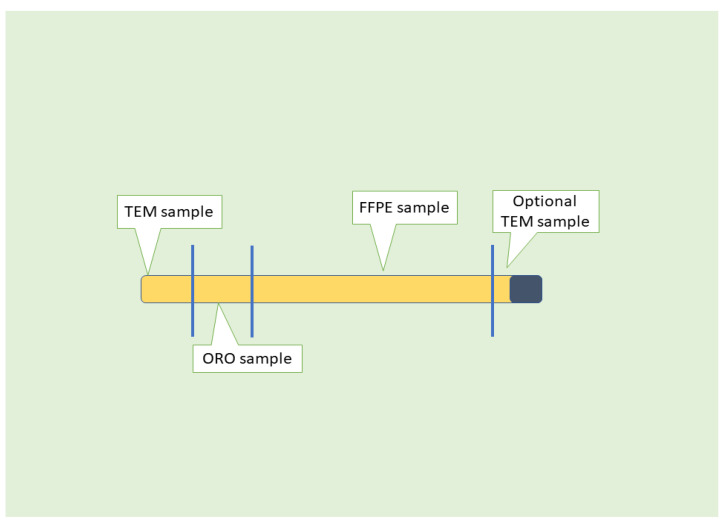
CHEO (Children’s Hospital of Eastern Ontario) Protocol for Pediatric Liver Biopsy. Schematic approach to a liver biopsy core with a sample reserved for FFPE, a sample dedicated to ORO staining (special stain) for cryostat cutting and staining for fat cell quantification, and one or two samples for ultrastructural examination (EM, electron microscopy). In case more cores are provided, the additional cores are channeled for formalin fixation and paraffin embedding (FFPE). Notes: TEM, transmission electron microscopy; FFPE, formalin fixation paraffin embedding; ORO, Oil red O.

**Figure 3 ijms-25-08462-f003:**
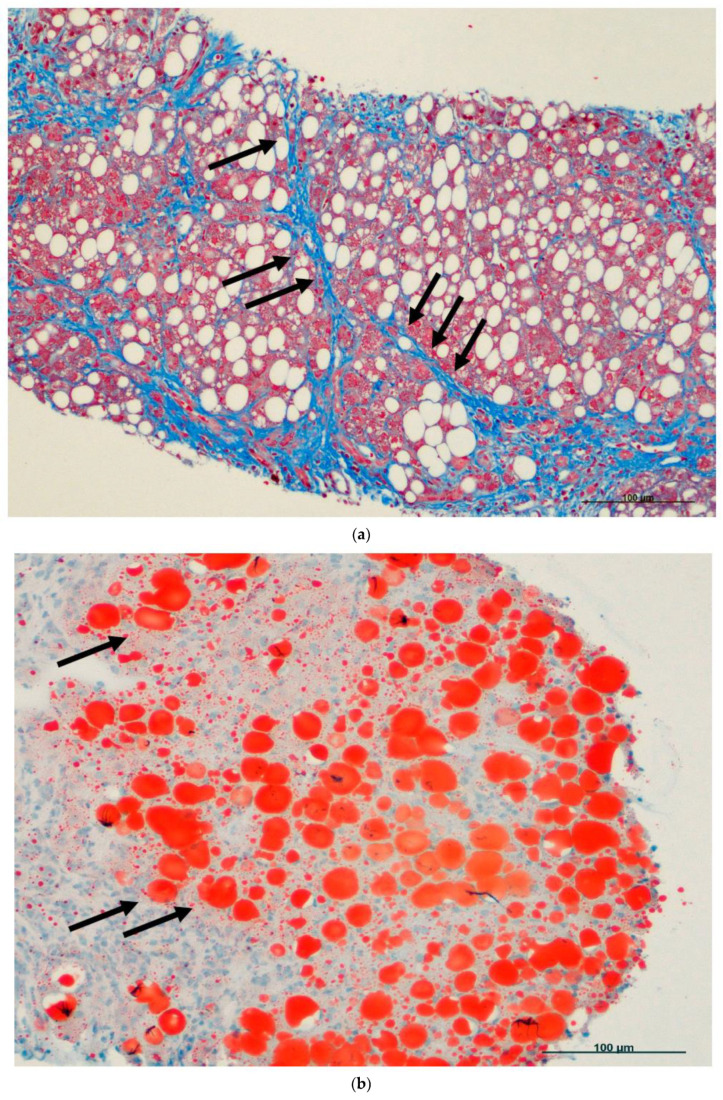
(**a**) Liver fibrosis. Extensive pericentral and periportal fibrosis with collagen deposition (blue) forming bridges and nearly pseudo-nodules (arrows). Masson’s trichromic stain, 100× original magnification, scale bar: 100 μm. (**b**) Liver steatosis. Fatty accumulation in the hepatocytes can be highlighted (orange) using the Oil red O staining (arrows; ORO stain, 100× original magnification, scale bar: 100 μm). (**c**) Hepatocytic ballooning. ‘Hepatocytic ballooning’ is an often-employed combined term in liver histology that denotes the degeneration of hepatocytes, characterized by their expansion, swelling, rounding, and the presence of reticulated cytoplasm (arrows; Hematoxylin–Eosin staining, 400× original magnification, scale bar: 10 μm). (**d**) Councilman bodies at portal and periportal areas. Councilman bodies (black arrows) are evidence of single-cell necrosis. A Councilman body, often referred to as a Councilman hyaline body or apoptotic body, is a pink-stained globule composed of pieces of dying liver cells. In the end, the fragments are engulfed by macrophages or nearby parenchymal cells. This pediatric patient was affected by Overlap syndrome, characterized by MASLD/MASH and autoimmune hepatitis, which is characterized by plasma cells (green arrow) infiltrating the portal tracts and evidence of interface hepatitis (Hematoxylin–Eosin staining, 400× original magnification, scale bar: 10 μm).

**Figure 4 ijms-25-08462-f004:**
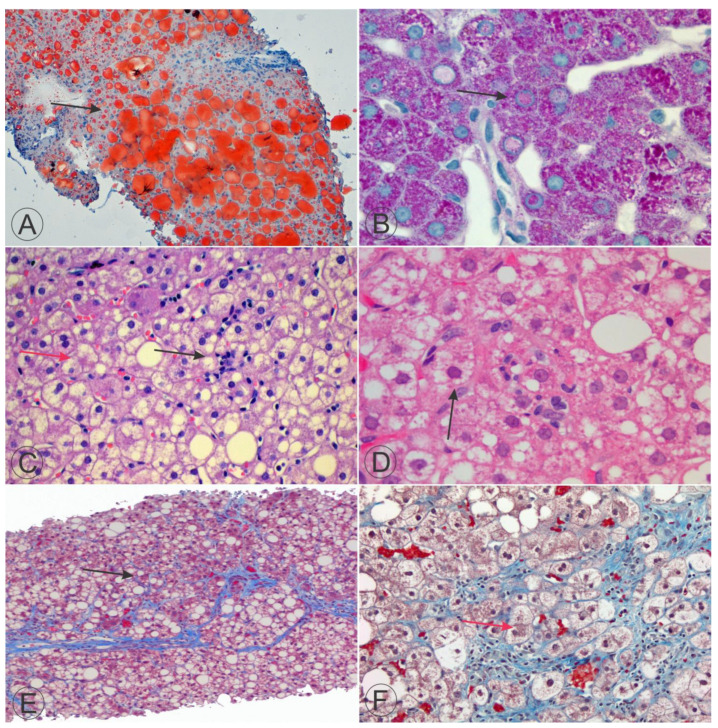
Single and multiple aspects of hepatocellular injury and portal tract damage. Upper microphotographs (**A**,**B**) show a low-power view (**A**) of two teenager patients with nonalcoholic fatty liver disease (NAFLD/MASLD) ((**A**), the arrow exquisitely exhibits the Oil-red-O stained vacuoles of the hepatocytes) and a high-power view (**B**) with Periodic acid Schiff (PAS)-stained nuclei in a pediatric patient with type 1 diabetes mellitus (T1DM) and Mauriac syndrome. Mauriac syndrome is a rare complication of T1DM. It is related to low-insulin concentrations and characterized by hepatomegaly, growth and puberty delay, as well as elevated transaminases and serum lipids. The middle microphotographs highlight lobulitis (black arrow) and ballooning (red arrow) in a child with nonalcoholic steatohepatitis (NASH) (**C**) as well as ballooning with a hepatocyte exhibiting a Mallory–Denk body (black arrow) (**D**). The lower microphotographs show heavy vacuolar degeneration and prominent bridging fibrosis and perisinusoidal fibrosis (“chicken wire”; black arrow) using a Masson’s trichromic stain (**E**) and prominent fibrosis and Malloy–Denk bodies (red arrow) using a slightly modified Masson’s trichromic to highlight perisinusoidal fibrosis (**F**). Masson’s trichrome is a three-color-staining procedure used in histology and different specific applications, but all are suited for distinguishing cells from surrounding connective tissue. Overall, the stain produces red keratin and muscle fibers, blue or green collagen and bone, light red or pink cytoplasm, and dark brown to black cell nuclei. The “slightly modified” Masson’s trichromic stain was achieved by increasing the time of exposure of the tissue with the solution C, also called fiber stain, which contains Light Green SF yellowish, or alternatively Fast Green FCF. ((**A**), Oil-red O stain, 100× original magnification; (**B**), PAS stain, 630× original magnification; (**C**), Hematoxylin and Eosin stain, 400× original magnification; (**D**), Hematoxylin and Eosin stain, 630× original magnification; (**E**), Masson’s trichromic stain, 100× original magnification; (**F**), “slightly modified” Masson’s trichromic stain, 200× original magnification). All microphotographs are from patients who presented at our clinics and were younger than 18 years of age at the time of the liver biopsy.

**Figure 5 ijms-25-08462-f005:**
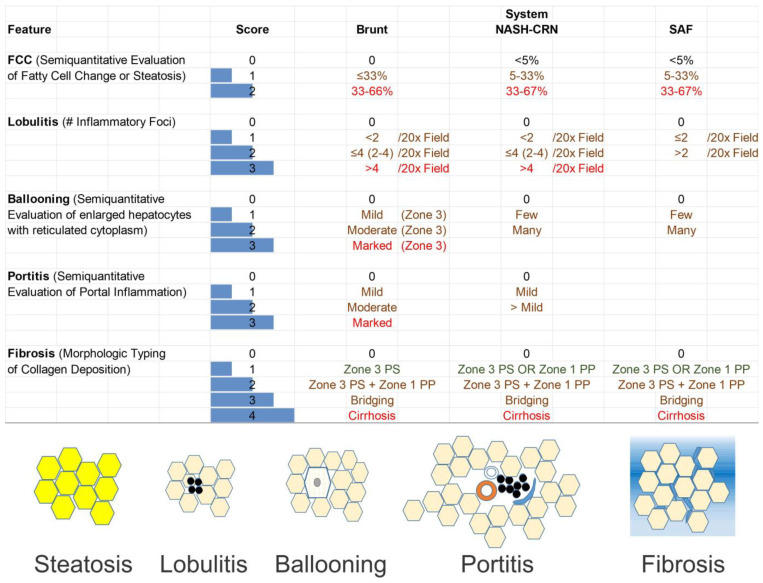
Comparison of the three most used nonalcoholic fatty liver disease/nonalcoholic steatohepatitis (NAFLD (MASLD)/NASH (MASH)) scoring systems. The three most commonly used scoring systems include the Brunt’s score, the NASH Clinical Research Network (NASH CRN), and the SAF (steatosis, activity, and fibrosis) score. The SAF score separates steatosis from parenchymal necroinflammation, which are two characteristics that may have distinct prognostic potential. Five features are scored in the Brunt’s and NASH CRN scores, while only four features are scored in the SAF score. The features are explained on the left side of the figure and cartoons are depicted at the base of the photograph.

**Table 1 ijms-25-08462-t001:** Conditions and/or diseases that can coexist with NAFLD/MASLD.

Obesity Insulin resistance, type 2 diabetes mellitus (T2DM) Hypercholesterolemia Dyslipidemia Hormonal deficiencies (growth hormone, thyroid hormones, and gonadotropins) Metabolic syndrome (MetS) Arterial hypertension Osteoporosis Obstructive sleep apnea syndrome (OSAS) Chronic kidney disease Psoriasis Cancer Psychiatric disorders

Notes: Metabolic syndrome (MetS) is defined as the condition of abdominal obesity, high blood pressure, low high-density lipoprotein cholesterol, hypertriglyceridemia, and hyperglycemia [3]. The list is probably not exhaustive of all conditions and/or diseases that can coexist with NAFLD/MASLD, and future updates will follow.

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
