# Peer review of "NAFLD (MASLD)/NASH (MASH): Does It Bother to Label at All? A Comprehensive Narrative Review"

_ijms, 2024, doi:10.3390/ijms25158462_

Round 1

Reviewer 1 Report

Comments and Suggestions for Authors

Based on a detailed review of the manuscript titled "NAFLD/NASH: Does it bother to label at all?", here are some comments and suggestions for revision:

  1. Abstract: The abstract should encompass research backgrounds, objectives, methodology (including research design, literature search and selection criteria, dependent variables, and quality assessment of prior studies), findings, conclusions, and recommendations.
  2. Introduction: While the introduction provides an adequate overview, it would benefit from a clearer statement of the manuscript’s objectives and structure. In line 40, replace "type II diabetes mellitus" with "type 2 diabetes mellitus" for consistency with standard terminology. Additionally, in line 44, clarify why liver biopsy is not well tolerated in children and discuss applicable alternative methods.
  3. Materials and Methodology: This review lacks explicit mention of research methods (e.g., research design, literature search and selection criteria, dependent variables, and quality assessment of prior studies). Detailed research methods should be explicitly described.
  4. Assessment of NAFLD/NASH using Liver Biopsy: Streamline lines 75-78 to focus on essential aspects of liver biopsy and its significance. In lines 86-89, use bullet points to list diseases that can coexist with NAFLD for improved readability. When explaining the liver biopsy procedure, ensure clarity, and consider briefly discussing the rationale behind specific biopsy techniques.
  5. Histopathology: Consider using diagrams or images to visually represent the histopathological features described (lines 92-137), enhancing comprehension. The discussion of histopathological features could be streamlined by grouping similar features together and employing bullet points or subheadings for clarity.
  6. AI in Pathology: Include recent studies or references supporting the use of AI in pathology (lines 138-157). Discuss current limitations and areas for improvement. While the section on AI is appropriately positioned, consider expanding it to address limitations and future directions in this field.
  7. Conclusion: Summarize key findings and their implications. Provide a clear statement on future directions or unanswered questions in NAFLD/NASH research.
  8. References: Ensure all references are correctly formatted according to the journal's guidelines. Inline citations should be consistently formatted.

Author Response

Abstract: The abstract should encompass research backgrounds, objectives, methodology (including research design, literature search and selection criteria, dependent variables, and quality assessment of prior studies), findings, conclusions, and recommendations.

Thank you for your comments and suggestions. The abstract now contains the required items. It is a comprehensive narrative review.
2.
Introduction: While the introduction provides an adequate overview, it would benefit from a clearer statement of the manuscript’s objectives and structure. In line 40, replace "type II diabetes mellitus" with "type 2 diabetes mellitus" for consistency with standard terminology. Additionally, in line 44, clarify why liver biopsy is not well tolerated in children and discuss applicable alternative methods.

Thank you for your comments and suggestions. The text has been corrected.

3.
Materials and Methodology: This review lacks explicit mention of research methods (e.g., research design, literature search and selection criteria, dependent variables, and quality assessment of prior studies). Detailed research methods should be explicitly described.

Yes, I did a comprehensive narrative review using a searching methodology according to a scoping review. I implemented the most updated guidelines for completeness only.

4.
Assessment of NAFLD/NASH using Liver Biopsy: Streamline lines 75-78 to focus on essential aspects of liver biopsy and its significance. In lines 86-89, use bullet points to list diseases that can coexist with NAFLD for improved readability. When explaining the liver biopsy procedure, ensure clarity, and consider briefly discussing the rationale behind specific biopsy techniques.

Thank you for your comments and suggestions. I have modified the text according to yours and the comments and suggestions received from the other reviewer, and I truly hope that the text is clearer than before. I also implemented Rinella's paper and the AASLD recommendations for the current terminology. More figures have been included in this revised manuscript.

5.
Histopathology: Consider using diagrams or images to visuallyrepresent the histopathological features described (lines 92-137), enhancing comprehension. The discussion ofhistopathological features could be streamlined by groupingsimilar features together and employing bullet points orsubheadings for clarity.

Yes, as indicated before I included more photographs and more arrows to delineate the changes seen in NAFLD (MASLD) / NASH (MASH)

6.
AI in Pathology: Include recent studies or references supporting the use of AI in pathology (lines 138-157). Discuss current limitations and areas for improvement. While the section on AI is appropriately positioned, consider expanding it to address limitations and future directions in this field.

This section has been expanded notably, and I have included the limitations I could gather from the current literature.

7.
Conclusion: Summarize key findings and their implications. Provide a clear statement on future directions or unanswered questions in NAFLD/NASH research.

The conclusive remarks have been revised as well, considering the comments and suggestions of both reviewers.

8.
References: Ensure all references are correctly formatted according to the journal's guidelines. Inline citations should be consistently formatted.

I included more references and formatted them using Endnote. Thank you again for your time on this project.

Reviewer 2 Report

Comments and Suggestions for Authors

This narrative review by Sergi highlights the assessment of NAFLD/MASLD using liver biopsy, the fatty cell change with or without inflammation, steatohepatitis, as well as grading and staging. 

It is a concise and well written review. This article gives a good overview of the topic and it is well structured. The figures and tables are clear and illustrative.

I suggest that the author adds a table with the most common causes of NAFLD in pediatric versus adult patients. Further I recommend to add a short paragraph discussing rare causes or predisposing risk factors of NAFLD, e.g. specific PNPLA3 gene variants.

Comments on the Quality of English Language

ok

Author Response

This narrative review by Sergi highlights the assessment of NAFLD/MASLD using liver biopsy, the fatty cell change with or without inflammation, steatohepatitis, and grading and staging.
It is a concise and well-written review. This article gives a good overview of the topic and is well structured. The figures and tables are clear and illustrative.
I suggest that the author add a table with the most common causes of NAFLD in pediatric versus adult patients. I also recommend adding a short paragraph discussing rare causes or predisposing risk factors of NAFLD, e.g., specific PNPLA3 gene variants.

Thank you for your comments and suggestions. I added a table with the most common causes of NAFLD/MASLD in pediatric versus adult patients. Moreover, I added a section on the predisposing genetic risk factors for NAFLD/MASLD. Specifically, I included a figure from Rinella's paper in Ann Hepatology, published in January of this year, concerning the most updated criteria of MASLD using the Creative Common Licence 4.0.

I hope that my revised manuscript may be considered suitable for publication.

Round 2

Reviewer 1 Report

Comments and Suggestions for Authors

The second draft of the manuscript has been appropriately revised based on the reviewer's comments. Thank you for your hard work.